# Increased Intracellular Free Zinc Has Pleiotropic Effects on Doxorubicin-Induced Cytotoxicity in hiPCS-CMs Cells

**DOI:** 10.3390/ijms24054518

**Published:** 2023-02-24

**Authors:** Kamil Rudolf, Emil Rudolf

**Affiliations:** Department of Medical biology and genetics, Faculty of Medicine in Hradec Kralove, Charles University, Zborovska 2089, 500 03 Hradec Kralove, Czech Republic

**Keywords:** hiPCS-CMs, doxorubicin, zinc, senescence, MAPK

## Abstract

(1) the mechanisms and outcomes of doxorubicin (DOX)-dependent toxicity upon changed intracellular zinc (Zn) concentrations in the cardiomyocytes obtained from human-induced pluripotent stem cells (hiPCS-CMs) were investigated; (2) cells exposed to the DOX were pretreated or cotreated with zinc pyrythione (ZnPyr) and various cellular endpoints and mechanisms were analyzed via cytometric methods; (3) both DOX concentrations (0.3 and 1 µM) induced a concentration-dependent loss of viability, an activation of autophagy, cell death, and the appearance of senescence. These phenotypes were preceded by an oxidative burst, DNA damage, and a loss of mitochondrial and lysosomal integrity. Furthermore, in DOX-treated cells, proinflammatory and stress kinase signaling (in particular, JNK and ERK) were upregulated upon the loss of free intracellular Zn pools. Increased free Zn concentrations proved to have both inhibitory and stimulatory effects on the investigated DOX-related molecular mechanisms, as well as on signaling pathways on the resulting cell fates; and (4) free intracellular Zn pools, their status, and their elevation might have, in a specific context, a pleiotropic impact upon DOX-dependent cardiotoxicity.

## 1. Introduction

Anthracycline compounds constitute a family of antibiotics of both bacterial and semisynthetic origin, with potent cytostatic and cytotoxic activities against a range of human malignancies [1]. Despite their clinical success within various therapeutic regimens, i.e., as monotherapy or in combination with other antineoplastic agents, a considerable number of the treated patients experience anthracycline-associated cardiotoxicity. This toxicity clinically presents in the form of left ventricular dysfunction, arrythmias, and later, dose-dependent cardiomyopathy, as well as cardiac failure [2,3]. One of the prominent anthracycline drugs which is often implicated in acute and chronic cardiotoxicity is doxorubicin (DOX). Earlier studies have suggested that DOX-induced cardiotoxicity and its resulting chronic heart failure are related to the DOX-mediated poisoning of topoisomerase II beta and/or to iron-catalyzed oxidative damage in cardiomyocytes [4,5]. More recent evidence argues, however, that DOX has multiple intracellular targets, thereby suggesting a more complex set of mechanisms behind its toxicity [6]. These include cardiomyocyte survival signaling, cardiac inflammation, energetic stress, and the general modulation of gene expression [7,8,9,10]. Furthermore, it is not clear to what extent DOX influences individual targets in exposed cardiomyocytes and how the signaling from these targets combines to produce the nature and/or spectrum of the resulting cell phenotypes, including demonstrated premature senescence, autophagy, or other regulated cell death modalities [11,12,13]. The elucidation of these points is of great relevance, since it may, for instance, help with the development and implementation of possible preventive measures, such as specific inhibitors with anticipated reduction, or the prevention of such undesirable, therapy-related consequences.

Zinc (Zn) is an essential micronutrient implicated in numerous biological processes within human cells, tissues, and organs [14]. The total Zn content of the human body is estimated to be in the range of grams, however, several tissues/cells are known to be extra rich in Zn, including the prostate, the skin, bones, and muscles [15]. While skeletal muscles account for an estimated 57% of the body’s total Zn content, the human heart stores approximately 0.4% of the body’s Zn [16]. In the cells, Zn exists both in labile and bound forms, whose concentrations, distribution, and availabilities are controlled with several influx, efflux, buffering, and muffling mechanisms [17]. Since Zn-dependent effects concern cell growth, gene expression, metabolism, signal transduction, and redox balance, as well as cell survival and cell death [18], any acute or chronic changes in the intracellular Zn levels are associated with redox stress, defective signaling, cell cycle disturbances, and an increased death rate, which clinically translate into defective growth and wound repair [19], neurodegeneration [20], and tumorigenesis [21]. In addition, the functional connection between disturbed Zn homeostasis, in particular Zn deficiency, and the development of cardiovascular diseases, is supported by several studies [22]. Concerning the fact that DOX-mediated stress and the damage of exposed cardiomyocytes involve multiple mechanisms and targets which are directly influenced by Zn, the manipulation of intracellular Zn levels to mitigate DOX-induced cardiotoxicity is of relevance, as supported by several published reports.

We have investigated the mechanisms and outcomes of DOX-dependent toxicity in cardiomyocytes obtained from human-induced pluripotent stem cells (hiPCS-CMs) in experimental conditions, mimicking the DOX dosing and timing used in the clinical context. We report the concentration-dependent DOX effects, including differential DNA damaging, an oxidative burst, and the activation of proinflammatory and stress kinase signaling, resulting in various endpoints including autophagy, cell death, and senescence. Moreover, the increased free Zn concentrations in DOX-exposed cells proved to have both inhibitory and stimulatory effects on particular molecular mechanisms that were investigated, as well as on the resulting cell fates.

## 2. Results

### 2.1. Viability and Cell Fates of iCell Cardiomyocytes Acutely or Chronically Exposed to DOX

hiPCS-CMs were exposed to 0.3 and 1 µM for 48 h, with a subsequent washout and monitoring for up to 6 days, to mimic both acute as well as delayed DOX toxicities over time. The viabilities of hiPCS-CMs (both control and DOX-exposed) were determined by a neutral red uptake assay (NRU). The viability of the control hiPCS-CMs gradually decreased in a time-dependent manner, and at the end of the study (192 h), reached approximately 60% (Figure 1A). Both of the employed DOX concentrations inhibited hiPCS-CMs viability, however, to different extents. In case of the 0.3 µM of DOX, the viability of the then-exposed cardiomyocytes was found at all of the analyzed time intervals to be lower than in the untreated control cells. On the other hand, the 1 µM of DOX induced a significant loss of hiPCS-CMs viability at 48 h of treatment, and this trend continued until the end of the experiment (Figure 1A).

The subsequent analyses aimed to verify the presence and the extent of the DOX-induced autophagy, cell death, and senescence in the exposed cells. In the hiPCS-CMs treated with both DOX concentrations (0.3 and 1 µM), an increased rate of autophagic flux, as well as the presence of autophagic vacuoles, were noted. In the lower DOX-concentration-exposed cells, significant levels of autophagy were reached in the time interval of 120–196 h only. Conversely, the higher employed DOX concentration had already induced significant autophagy at 24 h in the treated cells, whose rate further grew until 72 h of treatment. At later time intervals, however, the rate of the observed autophagy dropped and became comparable in both DOX treatment models (Figure 1B,C).

Similar trends were noted in the detected extent of DOX-induced cell death. Both DOX concentrations caused a significant cell death rate in the exposed hiPCS-CMs, which peaked at 72 h of treatment (6.2% with the 0.3 µM of the DOX and 21% with the 1 µM of the DOX), and later gradually (0.3 µM of DOX) or steeply decreased (1 µM of DOX) (Figure 1D). Further analyses confirmed that the predominant mode of DOX-associated cell demise in the treated cardiomyocytes was non-apoptotic.

DOX at both concentrations further produced a time-dependent accumulation of senescent phenotype, as evidenced by an increased β-galactosidase positivity in the treated cells, which reached significant levels at 72 h of treatment and continued to rise until the end of the experiment. Unlike autophagy and cell death, however, senescent phenotype in the studied cardiomyocytes was more significantly induced with a lower DOX concentration (Figure 1E).

### 2.2. Effects of Acute and Chronic DOX Exposure on DNA Damage, Oxidative Stress, Mitochondrial and Lysosomal Status in iCell Cardiomyocytes

The exposure of hiPCS-CMs to both of the DOX concentrations resulted in a time-dependent growth in DNA damage, coinciding with a similar increase in the production of the superoxide ion. Its levels culminated at 120 h of exposure in the exposed cardiomyocytes, and, unlike DNA damage, showed a marked concentration dependence (Figure 2A,C). The mitochondrial production of ATP in the exposed cardiomyocytes decreased in time too, whereas lysosomal membrane permeabilization reached its maximum at 48 h of exposure, and remained unchanged until the end of the experiment. In both instances, no difference in the extent of the mitochondrial ATP production, as well as lysosomal damage upon employing the DOX concentrations, was recorded (Figure 2B,D).

### 2.3. Effects of Acute and Chronic DOX Exposure on Intracellular Signaling in iCell Cardiomyocytes

DOX-induced changes in select signaling pathways of the hiPCS-CMs related to the altered intracellular targets were studied using several functional assays. These revealed that the treatment of cardiomyocytes with DOX enhanced the p53 activity in a time- and concentration-dependent manner (Figure 3A). In contrast, the NF-κB activity within the same treatment conditions indicated marked concentration differences. Thus, 1 µM of DOX induced a rapid, massive rise in NF-κB activity, culminating at 72 h of exposure with a gradual decline. Conversely, in the 0.3 µM DOX-exposed cells, the NF-κB activity increased very slowly, but never significantly (Figure 3B).

A lower DOX concentration had no effect on the p38 activity throughout the entire 192 h experiment. The JNK activity, on the other hand, first markedly grew between 72 h and 120 h of the DOX exposure, and decreased again at later time intervals. The ERK activity increased between 120 h and 192 h of the DOX treatment (Figure 3C). Similarly, no changes in the p38 activity were recorded in the cardiomyocytes exposed to a higher DOX concentration. Still, this concentration-induced rapid rise in both JNK (already at 24 h) and ERK (at 48 h) activities and their high levels persisted in the treated cells until the end of the experiment (192 h) (Figure 3D).

### 2.4. Effects of Acute and Chronic DOX Exposure on Free Zn Levels in iCell Cardiomyocytes

The free Zn levels in the control and DOX-exposed hiPCS-CMs were monitored during the entire course of the experiment (192 h). In the control cells, the detected free Zn gradually decreased in a time-dependent manner, and this decline corresponded with the observed gradual loss of cell viability (Figure 1A and Figure 4). The exposure of the cardiomyocytes to 0.3 µM of DOX induced a time-dependent loss of the intracellular free Zn content that reached significant levels at 120–192 h of treatment (Figure 4). The higher DOX concentration of 1 µM significantly depleted the free Zn content in the cardiomyocytes already at 48 h of exposure, and this trend continued until the end of the experiment (Figure 4).

### 2.5. Concentration-Dependent Effects of Zinc Pyrithione on Viability and on Free Zn Levels in iCell Cardiomyocytes

To increase the intracellular Zn content, Zn ionophore ZnPyr was used. Firstly, various ZnPyr concentrations were tested to verify their cytotoxicity. The ZnPyr concentrations higher than 0.25 µM had a significant cytotoxic effect on the treated cells (Figure 5A). On the other hand, the ZnPyr concentration of 0.1 µM had a slightly positive effect on the viability of the treated cardiomyocytes, and was thus used in further experiments (Figure 5A). The treatment of cardiomyocytes with the 0.1 µM of ZnPyr alone, or with 0.3 µM and 1 µM of DOX, could maintain the intracellular free Zn concentrations at relatively stable levels, except for those in the 1 µM DOX-exposed cells at 192 h (Figure 5B).

### 2.6. Viability and Cell Fates of iCell Cardiomyocytes with Increased Intracellular Free Zn Content and Acutely or Chronically Exposed to DOX

The viability and other measured endpoints of the cardiomyocytes with enhanced intracellular free Zn content were next determined using previously described approaches. The presence of enhanced free Zn levels generally increased the viability of the control and the DOX-exposed (both concentrations) cardiomyocytes, although this increase reached significant values in the DOX-treated cells only. Moreover, the viability of the cells treated with 1 µM of DOX, however markedly elevated, nevertheless remained significantly lower when compared to the untreated cardiomyocytes (Figure 6A). Increased intracellular Zn levels had no effect on the autophagy rates of the untreated or DOX-treated cells (Figure 6B,C). The ZnPyr-mediated influx of Zn in the cardiomycytes also reduced the rate of DOX-related cell death, albeit in the case of the 1 µM of the DOX concentration only (Figure 6D). In addition, an enhanced intracellular free Zn content showed opposing effects on the cellular senescence. It reduced the presence of cells with this phenotype in the untreated and 0.3 µM DOX-treated cardiomyocytes, while increasing it in the 1 µM DOX-exposed cells (Figure 6E).

### 2.7. DNA Damage, Oxidative Stress, and Mitochondrial and Lysosomal Status in iCell Cardiomyocytes with Increased Intracellular Free Zn Content and Acutely or Chronically Exposed to DOX

Increased free Zn levels had no significant effect on DOX-induced (both concentrations) DNA damage in the exposed cardiomyocytes (Figure 7A). On the other hand, in the same cells, the mitochondrial ATP production, as well as the lysosomal membrane integrity, were markedly improved. A similarly significant reduction in the superoxide ion production, irrespective of the employed DOX concentration, was observed too (Figure 7B–D).

### 2.8. Intracellular Signaling in iCell Cardiomyocytes with Increased Intracellular Free Zn Content and Acutely or Chronically Exposed to DOX

While the increased presence of free intracellular Zn levels had no decisive effect on DNA-p53 binding in the hiPCS-CMs exposed to both DOX concentrations, in the same cells, it clearly reduced the NF-κB activity, in particular upon their exposure to 1 µM of DOX (Figure 8A,B). In addition, intracellular-Zn-enriched cardiomyocytes displayed varying activities of the following MAPKs related to the employed DOX concentrations. Thus, in the case of the 0.3 µM of DOX, a significant decrease in the JNK activity occurred, while in 1 µM DOX-treated cells, both the JNK and ERK activities decreased, however, this decrease was significant in the case of ERK only (Figure 8C,D).

## 3. Discussion

Anthracycline-, and, in particular, DOX-dependent cardiotoxicity (sometimes referred to as type I cardiotoxicity) is caused by cardiomyocyte death, which leads to a permanent injury and an irreversibly changed physiological status of the heart [3]. Initially, DOX-induced cell damage and their subsequent death have been attributed to its inhibitory effect towards topoisomerase II beta and/or its ability to induce an iron-catalyzed oxidative burst in exposed cells. Today, however, it is believed that DOX-related adverse effects in cardiomyocytes are more complex, since DOX may affect multiple intracellular targets, i.e., including various membrane receptors, as well as intracellular signaling nodes and pathways [6,23]. It has been reported that DOX may simultaneously trigger different regulated cell death modalities, as well as other endpoints, including apoptosis, senescence, autophagy, necroptosis, or pyroptosis [12,13,24]. Accordingly, the use of various specific inhibitors (with the exception of iron-chelators) failed to reduce cardiotoxicity in the clinic [6], thereby highlighting the pleiotropy of the DOX-influenced targets and responses, as well as the need to identify new molecular players for a better understanding of the mechanisms of such a process.

The present study featured an in vitro model of acute and delayed DOX toxicities in hiPCS-CMs. The cells were exposed to DOX concentrations of 0.3 µM and 1 µM for 48 h, with a subsequent washout and monitoring for up to 6 days to mimic clinical time and concentration exposures [25,26]. Our results demonstrated that the employed DOX concentrations induced several endpoint phenotypes in cardiomyocytes. The 1 µM of DOX mainly induced autophagy and chiefly non-apoptotic cell death (with a maximum at 72 h of exposure), whereas the cells with the senescent phenotype started to appear only gradually, from 72 h onwards. Conversely, the lower DOX concentration (0.3 µM) induced significantly lower cell death rates, while gradually stimulating the autophagy rate and senescence. The DOX-mediated autophagy in the present model may appear as surprising in light of the recently published evidence. In a model mouse heart, it was demonstrated that DOX blocks autophagic flux by impairing lysosomal function, and its specific cardiotoxicity has been chiefly attributed to this mechanism [27]. Our opposing evidence may be partially attributed to the differences in the used models, as well as in the setup of the experiments, however, there are very likely other factors involved. One of them might be lysosomes, whose specific impairment could affect the rate of autophagy in the context of time-dependent DNA damage, oxidative stress, and the loss of mitochondrial ATP production (Figure 2), as has been observed in other in vitro and in vivo settings [28,29,30]. Given the newly proposed effect of DOX on the several signaling networks in the cardiomyocytes which control DNA damage response, survival, and inflammation, among others, we also checked the activities of TP53 and NF-κB, as well as selected MAPKs—JNK, ERK, and p38 in the present model [31,32,33]. Our data show the involvement of all of these targets within DOX treatment, although with a concentration- and time-dependent relationship. Thus, predictably, a higher DOX concentration induced early DNA damage and proinflammatory signaling, accompanied by increased activities of ERK and JNK. Conversely, a lower DOX concentration had a more gradual stimulatory effect on these targets. The important role of individual MAPKs (as compared with other mechanisms) in the DOX-induced senescent phenotype in the present model was also independently confirmed by the use of specific pharmacological inhibitors. This finding is of particular interest, since in H9c2 rat cardiomyocytes, DOX-induced senescence is promoted by the expression of the DNA damage response-1 (Redd1) gene acting as a downstream effector of p38 and proinflammatory signaling [34]. In addition, in cord blood endothelial progenitor cells, activated p38 is known to induce senescence, while JNK attenuates it [35]. Here, our alternative observation may be explained by the nature of the differences in the used model and experimental setting, including the time aspect. Another factor to consider is an apparent complexity of the biological response itself, since evidence from other published reports further expands a list of putative mechanisms behind DOX-induced senescence. These include a telomere binding factor 2 modulating androgen receptor pathway [36], a ligand-activated transcriptional factor PPARδ [37], as well as individual components of the cardiac microenvironment [38]. Due to the fact that DOX-driven senescence in cardiomyocytes, next to other cell fate modalities, appears to be fundamental to the final clinical outcome, further comparative analyses on relevant models are needed to reveal the critical players and targets of possible future interventions.

Zinc is a microelement whose physiological roles in various organs, including the heart, are quite essential, as demonstrated by reported cases of Zn deficiency [22]. Moreover, supplementation with Zn helps to protect cells against a range of stressors, including DOX, via Zn-induced metallothionein expression [39]. Since the treatment with DOX resulted in a time-dependent loss of free Zn pools in the hiPCS-CMs, we sought to address this by using an established zinc ionophore ZnPyr at the cell-tolerated concentration of 0.1 µM. The free Zn enrichment significantly improved the viability of the DOX-treated cells, however, with no effect on autophagy and with concentration-dependent effects on the cell death and senescence rates. Predictably, Zn significantly reduced the superoxide production, restored the mitochondrial performance, and prevented lysosomal membrane permeabilization, which corresponds to its known role as an antioxidant [40]. Moreover, the fact that the rate of DNA damage and related TP53 activity decreased only slightly confirms that DOX-induced damaging in cardiomyocytes occurs via multiple mechanisms. The other observed Zn-related inhibitory effects (i.e., proinflammatory signaling and MAPK activities) differed in relation to the used DOX concentration. Again, and interestingly, enhanced free Zn pools reduced the JNK activity in the cardiomyocytes treated with 0.3 µM of DOX, whereas with 1 µM of DOX, a similar inhibition concerned ERK activity. It would seem to imply that some DOX-influenced intracellular targets and responses are activated irrespective of the DOX concentration used (typically DNA-damage-associated), whereas others (MAPKs) operate in a concentration-specific manner. We believe that such a finding is of great importance, namely in the context of DOX-induced senescence, for instance, since enhanced intracellular free Zn content may both increase or decrease the senescence rates via the apparently distinct MAPKs.

In this work, we demonstrated a range of the dose-dependent effects of DOX on exposed human cardiomyocytes. Additionally, we showed that elevated free Zn content in these cells has pleiotropic effects on DOX toxicities, both reducing the resulting death rate, as well as inducing the senescence. Based on these results, we argue that Zn may have a beneficial role in DOX-dependent cardiotoxicity, however, this potential benefit is strongly context dependent. At this point, it is also necessary to mention the several limitations of the present study. These concern the biological limits of the used model, which does not allow the study to extend beyond the life span of cardiomyocytes. Secondly, and most importantly, the findings in this study are based on just one human cell line, representing just one cell type. Thus, the robustness of the obtained data is not optimal to allow for the drawing of more generalized conclusions, and would therefore require other models to be used to confirm the reported results. The current study did not take into consideration the role of metallothioneins as major Zn-binding, inducible systems that play several protective roles in the cells. The reason for this omission was the treatment scheme with a simultaneous direct supply of the Zn to the cardiomyocytes via ZnPyr, as well as with DOX. When treated this way, the cells had no time to induce metallothionein synthesis in the same way it is seen in classical Zn pretreatment experiments.

## 4. Materials and Methods

### 4.1. Cell Cultivation, Plating and Maintenance

Prior to the manipulation with cells, 96-well plates were pretreated with a solution of fibronectin (1 µg/mL, 600 µL/well) for 24 h at 37 °C and 5% CO_2_. hiPCS-CMs (Cellular Dynamics International, Madison, WI, USA) in cryovials were thawed and resuspended in plating medium, and their viability and density were determined with a hemocytometer (the viability rate was beyond 85%). Cells (30,000 cells/well) were plated into thus-prepared 96-well plates and maintained at 37 °C and 5% CO_2_ for 48 h. Next, plating medium in each well was exchanged for maintenance medium, which was thereafter typically exchanged every 48 h, as per the recommendation of the producer, until the end of the experiment and/or existence of the cultivation.

### 4.2. Experimental Scheme

The hiPCS-CMs in the 96-well plates were maintained in fresh DOX-free maintenance medium, with or without zinc pyrithione (ZnPyr) (controls), or in maintenance medium with dissolved DOX at the tested concentrations (0.3 and 1 µM), again, with or without ZnPyr. Generally, iCell cardiomyocytes were exposed to DOX for 48 h, then the medium with the DOX was removed, and the cells in the DOX-free maintenance medium were analyzed for another 6 days using several assays.

### 4.3. Chemicals

The zinc pyrithione, horseradish peroxidase, Triton-X, propidium iodide, dithiotreitol (DTT), JC-1, acridine orange, doxorubicin, monodansylcadaverine (MDC), Newport Green diacetate, 4′,6-Diamidine-2′-phenylindole dihydrochloride (DAPI) and neutral red were obtained from Sigma-Aldrich (St. Louis, MO, USA). The monoclonal rabbit anti-phospho Ser-139 H2A.X was from Cell Signaling Technology, Inc. (Danvers, MA, USA). The secondary antibodies were from Alexis Corporation (Lausen, Switzerland) and Dako (Glostrup, Denmark). The MitoSOX™ Red was from ThermoFisher Scientific Inc. (Carlsbad, CA, USA). All of the other chemicals were of the highest analytical grade.

### 4.4. Cell Viability

The control and DOX-treated hiPCS-CMs (with or without ZnPyr) in the 96-well plates were, at the particular time intervals, washed with fresh maintenance medium (the old medium was removed). Next, the cells were washed twice with PBS, and a medium with 100 μL of neutral red (80 µg/mL) in fresh medium was added to each well for 3 h (37 °C, 5% CO_2_). Thereafter, the medium was removed and the cells were exposed to 100 l of fixative solution (calcium chloride in 0.5% formaldehyde). The cells were further incubated for 15 min at RT. After two subsequent rounds of washing, 200 l of lysis solution (1% acetic acid in 50% ethanol) was added. After 30 min of incubation, the color product was measured spectrophotometrically at 540 nm, with 620 nm of reference wavelength (TECAN SpectraFluor Plus (TECAN Austria GmbH, Grödig, Austria). In all cases, the absorbance of the tested doxorubicin in medium alone was recorded to correct for potential interference.

### 4.5. Cell Damage and Death

The morphologies of the control and DOX-exposed hiPCS-CMs (with or without ZnPyr) were, at the specified time points, analyzed by the Cell Scoring Module of the MetaXpress^®^ Image Acquisition and Analysis Software (version 4), with a minimum of 5000 cells (per given time interval) evaluated. In addition, the nuclear architecture was determined by specific fluorescent staining, using DAPI with a subsequent microscopic evaluation. A minimum of 5000 cells (per given time interval) were evaluated.

### 4.6. Autophagy

Autophagy in the control and DOX-exposed hiPCS-CMs (with or without ZnPyr) was measured by MDC specific fluorescence, whose results were further validated with the measurement of 2 autophagy flux rates. The cells grown in 96-well plates with a transparent bottom were incubated with 50 µM of MDC in DMSO for 15 min at 37 °C in the dark. Thereafter, the wells were thoroughly rinsed twice in PBS, and MDC-positive signals were quantified using a SpectrafluorPlus (TECAN Austria GmbH, Grödig, Austria) at 390 nm (excitation) and 455 nm (emission). The results are shown as a relative intensity of the MDC-specific fluorescence in arbitrary units.

The autophagy flux was detected using the Premo™ Autophagy Tandem Sensor RFP-GFP-LC3B Kit (Thermo Fischer Scientific, NY, USA). The results were expressed as the ratio of red-specific fluorescence to yellow-specific fluorescence puncta per cell. In total, 500 cells per coverslip were analyzed. The samples were done in triplicates.

### 4.7. Senescence

The control and DOX-exposed hiPCS-CMs (with or without ZnPyr) in 96-well plates were washed with PBS, fixed with 2% formaldehyde, and incubated overnight (37 °C) with a freshly prepared SA-β-gal staining solution containing 1 mg/mL 5-bromo-4-chloro-3-indolyl β-D-galactopyranoside (X-gal) (Calbiochem, EMD Biosciences, Inc., La Jolla, CA, USA), 5 mM potassium ferrocyanide, 5 mM potassium ferricyanide, 150 mM NaCl, 2 mM MgCl_2_, and 40 mM citric acid, titrated to pH 6.0. Next, the specimens were washed with distilled H_2_O, dehydrated, and the SA-β-gal positivity was determined with the Cell Scoring Module of the MetaXpress^®^ Image Acquisition and Analysis Software. The results were expressed as the percentage of senescent cells.

### 4.8. Intracellular Free Zinc Content

The free intracellular zinc levels in the assayed control and DOX-exposed hiPCS-CMs (with or without ZnPyr) were determined fluorimetrically. The cells grown in the black-bottom 96-well plates were incubated with Newport Green diacetate (5 μM/mL in PBS, in the dark, for 30 min at 37 °C). The fluorescence intensity was determined by the multiplate reader TECAN SpectraFluor Plus (TECAN Austria GmbH, Grödig, Austria). The results in relative light units were obtained from the raw data minus the reagent blank, with the changes expressed as a percentage of controls.

### 4.9. DNA Damage

The control and DOX-exposed hiPCS-CMs (with or without ZnPyr) in the 96-black bottom microplates were fixed by formadehyde (10%, 30 min, 20 °C) and the expression of H2A.X in the cell nuclei was revealed via indirect immunodetection with a primary (1: 200, 60 min, 4 °C) and secondary (1:250, 60 min, 25 °C) antibody system, measured in a multiplate reader TECAN SpectraFluo Plus (TECAN Austria GmbH, Grödig, Austria) at a 485/520 nm filter combination. The results were expressed as an increase in the fluorescence intensity compared with the control cells.

### 4.10. Superoxide Production

The control and DOX-exposed hiPCS-CMs (with or without ZnPyr) grown in the 96-well plates with a black bottom were incubated with MitoSOX™ Red solution (5 µM, 20 min, 37 °C), rinsed in warm medium, and the specific fluorescence-reflecting superoxide was analyzed by the Cell Scoring Module of the MetaXpress^®^ Image Acquisition and Analysis Software. The results were expressed as the percentage of cells positive for the superoxide ion.

### 4.11. Mitochondrial Membrane Potential (ψm)

The control and DOX-exposed hiPCS-CMs (with or without ZnPyr) were stained with cationic JC-1 dye (10 μg/mL, 15 min, 37 °C) and the changes in the mitochondrial membrane potential were analyzed by the Cell Scoring Module of the MetaXpress^®^ Image Acquisition and Analysis Software.

### 4.12. ATP Production

The ATP content in the control and DOX-exposed hiPCS-CMs (with or without ZnPyr) was measured by an ATP bioluminescent assay kit (Sigma-Aldrich, Cat. No. FLAA, Prague, Czech Republic), as recommended by the manufacturer. The results were expressed as a percentage of control.

### 4.13. Lysosomal Membrane Assay

The control and DOX-exposed hiPCS-CMs (with or without ZnPyr) were stained with acridine orange (5 μM, 15 min, 37 °C), and its redistribution was measured fluorimetrically (TECAN SpectraFluorPlus, TECAN Austria GmbH, Grödig, Austria). Lysosomal membrane damage was expressed as an increase in diffuse cytosolic green fluorescence by the acridine orange released from lysosomes in arbitrary units.

### 4.14. TP53 DNA-Binding Assay

The TP53 DNA-binding activity in the control and DOX-exposed hiPCS-CMs (with or without ZnPyr) was measured using a p53 transcription factor assay kit (Cayman Europe, Tallinn, Estonia), as per the instructions of the manufacturer. The DNA binding that reflected the transcription activity of the TP53 in the cultures was determined spectrophotometrically at 450 nm, and expressed as percentage of control.

### 4.15. NF-κB Activation

The NF-κB activity (the binding of NF-κB p65 to DNA) in the control and DOX-exposed hiPCS-CMs (with or without ZnPyr) was measured with an NF-κB p65 transcription factor assay kit (Abcam, Cambridge, UK), according to the user manual. The absorbance of the samples reflecting the NF-κB DNA-binding activity was measured spectrophotometrically at 450 nm.

### 4.16. MAPK Activities

The whole cell extracts of the control and DOX-exposed hiPCS-CMs (with or without ZnPyr) were prepared by lysis in a cell extraction buffer (10 mM Tris, 100 mM NaCl, 1 mM EDTA, 1 mM EGTA, 1 mM NaF, 20 mM Na_4_P_2_O_7_, 2 mM Na_3_VO_4_, 1% Triton X-100, 10% glycerol, 0.1% SDS, and 0.5% deoxycholate, with a 1 mM protease inhibitor cocktail) for 30 min, on ice, with vortexing at 10 min intervals. The ERK, p38, and c-Jun N-terminal kinase (JNK) activities were measured using ELISA kits (Sigma-Aldrich, St. Luis, MO, USA and Calbiochem, San Diego, CA, USA), specific for total ERK and phospho-ERK (pTpY^185/187^), total and phospho-p38 (pTpY^180/182^), and total and phospho-JNK (pTpY^183/183^), according to the manufacturer’s instructions. The samples were read against the standard curves obtained from ERK and phospho-ERK, p38 and phospho-p38, and JNK and phospho-JNK standards. The results were normalized to micrograms of protein in the cell extract and expressed as the ratio of phospho to total kinase in the same sample.

### 4.17. Statistical Analysis

All of the experiments were repeated at least three times. The data analysis was performed using GraphPad Prism (GraphPad Software version 6.0, Inc., San Diego, CA, USA). The statistical analysis was carried out using one-way analysis of variance (ANOVA), followed by Dunnett’s multiple comparisons test, significant at a level of *p* < 0.05.

## Figures and Tables

**Figure 1 ijms-24-04518-f001:**
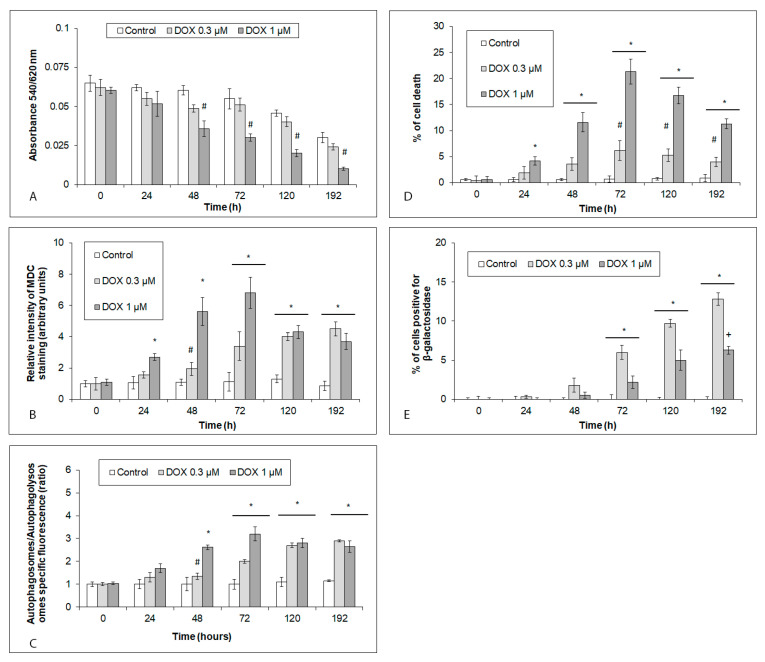
Effects of DOX (0.3 and 1 µM) on the viability, autophagy, cell death, and senescence of hiPCS-CMs during 192 h, as measured by particular assays described in Section 4. (**A**) Viability, (**B**,**C**) autophagy, (**D**) cell death, and (**E**) senescence. Values represent means ± SD of N = 3 (viability), N = 4 (autophagy), N = 3 (cell death), and N = 3 (senescence) experiments. * *p* < 0.05 compared to untreated control cells at the same treatment interval; # *p* < 0.05 compared to 1 µM DOX treated cells at the same treatment interval; + *p* < 0.05 compared to 0.3 µM DOX treated cells at the same treatment interval with one-way ANOVA test and Dunnett’s post test for multiple comparisons.

**Figure 2 ijms-24-04518-f002:**
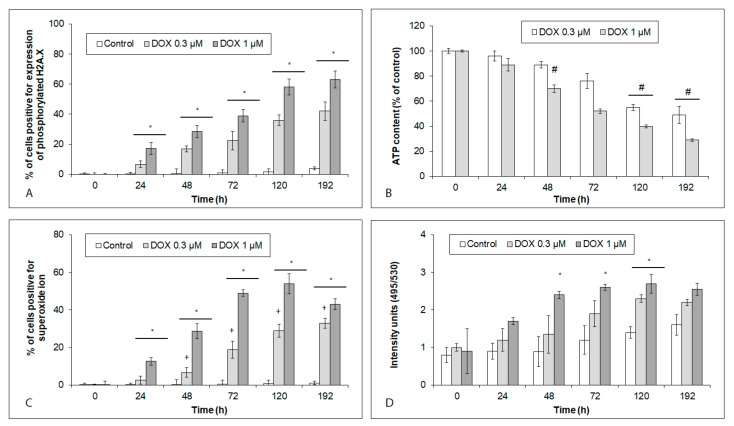
Effects of 0.3 µM and 1 µM DOX on various intracellular targets in hiPCS-CMs during 192 h. Control and DOX-exposed cardiomyocytes were harvested at individual time intervals, and (**A**) DNA damage, (**B**) mitochondrial production of ATP, (**C**) generation of superoxide ion, and (**D**) lysosomal membrane permeabilization, were determined by particular assays and procedures, as described in Section 4. Values represent means ± SD of N = 3 (DNA damage), N = 3 (ATP production), N = 3 (superoxide), and N = 3 (lysosomal permabilization) experiments. * *p* < 0.05 compared to untreated control cells at the same treatment interval; # *p* < 0.05 compared to untreated control cells at the same treatment interval; and + *p* < 0.05 compared to 1 µM DOX treated cells at the same treatment interval with one-way ANOVA test and Dunnett’s post test for multiple comparisons.

**Figure 3 ijms-24-04518-f003:**
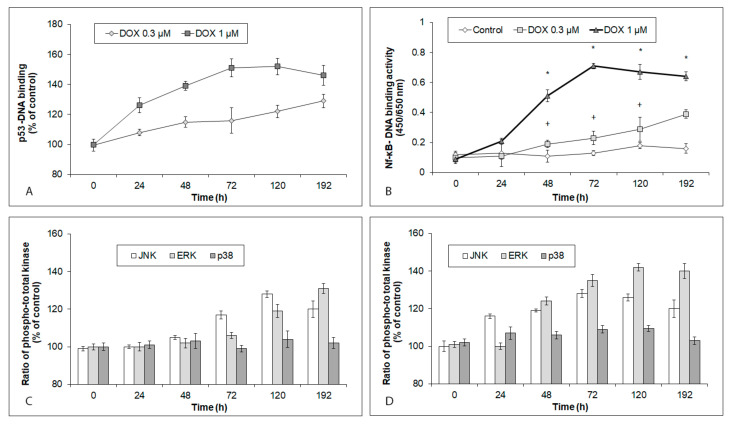
Effects of 0.3 µM and 1 µM DOX on intracellular signaling in hiPCS-CMs during 192 h. Control and DOX-exposed cardiomyocytes were harvested at individual time intervals, and (**A**) p53 activity, (**B**) NF-κB activity, (**C**) MAP kinases activity upon treatment with 0.3 µM DOX, and (**D**) MAP kinases activity upon treatment with 1 µM DOX, were determined by particular assays and procedures, as described in Section 4. Values represent means ± SD of N = 3 (p53 activity), N = 3 (NF-κB activity), N = 3 (MAP kinases activity at 0.3 µM DOX), and N = 3 (MAP kinases activity at 1 µM DOX) experiments. * *p* < 0.05 compared to untreated control cells at the same treatment interval; and + *p* < 0.05 compared to 1 µM DOX treated cells at the same treatment interval with one-way ANOVA test and Dunnett’s post test for multiple comparisons.

**Figure 4 ijms-24-04518-f004:**
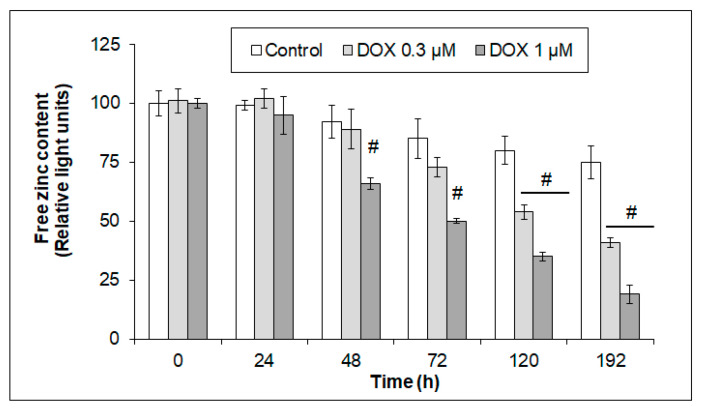
Effects of 0.3 µM and 1 µM DOX on intracellular free Zn levels in hiPCS-CMs during 192 h. Free zinc content in control and DOX-treated cells was measured by microfluorometry of the zinc-specific dye Newport Green diacetate, as described in Section 4. Values represent means ± SD of N = 4 experiments. # *p* < 0.05 compared to untreated control cells at the same treatment interval with one-way ANOVA test and Dunnett’s post test for multiple comparisons.

**Figure 5 ijms-24-04518-f005:**
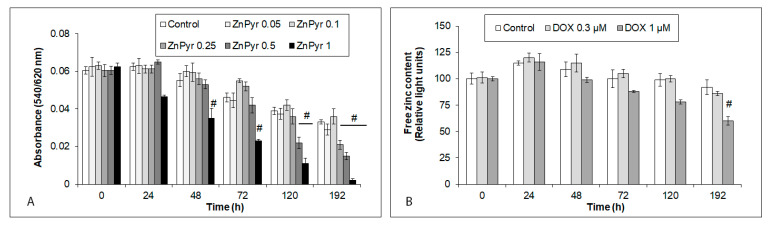
Effects of zinc pyrithione (ZnPyr) on viability and on free Zn levels in hiPCS-CMs over 192 h. Cardiomyocytes were exposed to a range of ZnPyr concentrations (in µM) and their viability was measured by neutral red uptake assay (NRU) (Section 4). Free zinc content in control and 0.3 µM and 1 µM DOX-treated cells was measured by microfluorometry of the zinc-specific dye Newport Green diacetate, as described in Section 4. Values represent means ± SD of N = 4 experiments. # *p* < 0.05 compared to untreated control cells at the same treatment interval with one-way ANOVA test and Dunnett’s post test for multiple comparisons.

**Figure 6 ijms-24-04518-f006:**
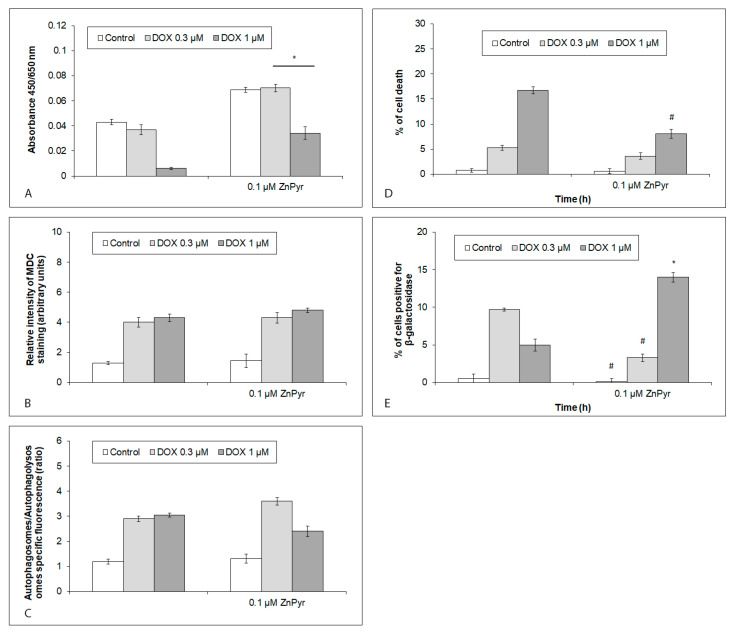
Effects of enhanced intracellular free Zn levels on viability and cell phenotypes in hiPCS-CMs exposed to 0.3 µM and 1 µM DOX at 120 h. Cardiomyocytes were treated with 0.1 µM zinc pyrthione (ZnPyr) alone or together with 0.3 µM and 1 µM DOX concentrations, and their (**A**) viability, (**B**,**C**) autophagy, (**D**) cell death, and (**E**) senescence were measured by particular assays (Section 4). Values represent means ± SD of N = 3 (viability), N = 4 (autophagy), N = 3 (cell death), and N = 3 (senescence) experiments. #,* *p* < 0.05 compared to untreated cells with enhanced free Zn content at the same treatment interval with one-way ANOVA test and Dunnett’s post test for multiple comparisons.

**Figure 7 ijms-24-04518-f007:**
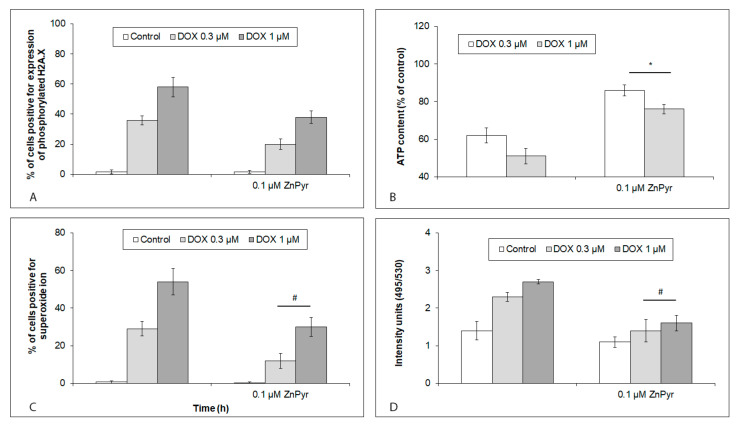
Effects of enhanced intracellular free Zn content on DNA damage, oxidative stress, and mitochondrial and lysosomal status in hiPCS-CMs exposed to 0.3 µM and 1 µM DOX at 120 h. Cardiomyocytes were treated with 0.1 µM zinc pyrthione (ZnPyr) alone or together with 0.3 µM and 1 µM DOX concentrations, and (**A**) DNA damage, (**B**) mitochondrial production of ATP, (**C**) generation of superoxide ion, and (**D**) lysosomal membrane permeabilization were determined by particular assays and procedures, as described in Section 4. Values represent means ± SD of N = 3 (DNA damage), N = 3 (ATP production), N = 3 (superoxide), and N = 3 (lysosomal permabilization) experiments. #, * *p* < 0.05 compared to untreated cells with enhanced free Zn content at the same treatment interval with one-way ANOVA test and Dunnett’s post test for multiple comparisons.

**Figure 8 ijms-24-04518-f008:**
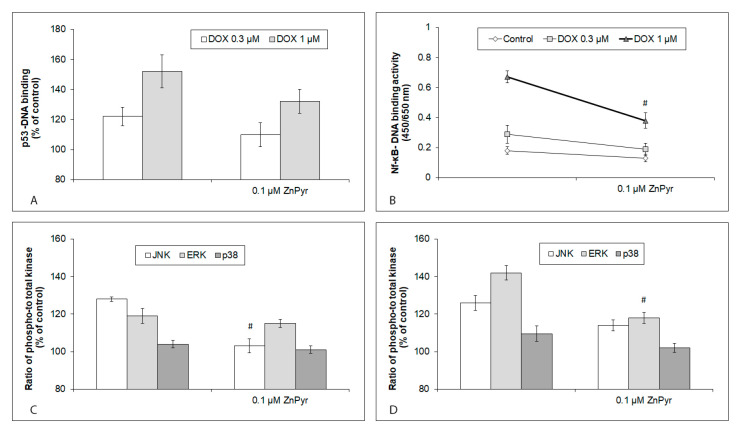
Effects of enhanced intracellular free Zn levels on intracellular signaling in hiPCS-CMs exposed to 0.3 µM and 1 µM DOX at 120 h. Cardiomyocytes were treated with 0.1 µM zinc pyrthione (ZnPyr) alone or together with 0.3 µM and 1 µM DOX concentrations, and (**A**) p53 activity, (**B**) NF-κB activity, (**C**) MAP kinases activity upon treatment with 0.3 µM DOX, and (**D**) MAP kinases activity upon treatment with 1 µM DOX were determined by particular assays and procedures, as described in Section 4. Values represent means ± SD of N = 3 (p53 activity), N = 3 (NF-κB activity), N = 3 (MAP kinases activity at 0.3 µM DOX), and N = 3 (MAP kinases activity at 1 µM DOX) experiments. # *p* < 0.05 compared to DOX treated cells at the same treatment interval with one-way ANOVA test and Dunnett’s post test for multiple comparisons.

## Data Availability

Data is contained within the article.

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
