# Peer review of "Increased Intracellular Free Zinc Has Pleiotropic Effects on Doxorubicin-Induced Cytotoxicity in hiPCS-CMs Cells"

_ijms, 2023, doi:10.3390/ijms24054518_

Round 1

Reviewer 1 Report

Manuscript ID # ijms-2104161

Title: "Increased intracellular free zinc has pleiotropic effects on doxorubicin-induced cytotoxicity in hiPCS-CMs cells"

This manuscript submitted by Rudolf K. and Rudolf E., is an original article regarding cell effect of doxorubicin on cardiomyocytes derived from human induced pluripotent stem cells (hiPSC-CMs). However the manuscript lacks an important set of experiments that validates the model. The authors should have assayed the differentiation of iPSC into cardiomyocytes by revealing the expression of specific biomarker with fluorescent microscopy. Without this initial control, there is no certainty that the cytotoxicity of doxorubicin was investigated on cardiomyocytes.

Furthermore, the WST-1 assay used to measure cell viability wasn’t appropriate because it is known that doxorubicin affects mitochondrial functions and the assay is based on the production of formazan by a mitochondrial dehydrogenase. So the cell viability must have been quantified with another assay.

As minor points:

-       There is no information in the manuscript about the method used for cell dead measurements.

-       The authors should mention for each experiment, the N (or number of replicates).

Author Response

Generally, we would like to thank the reviewer for her/his comments, constructive criticism and suggestions to improve the quality of our present work. All the raised questions and comments were taken into consideration and requested changes were implemented both as a part of originally existing evidence as well as based on the newly acquired data. Also, the implementation of new data as well as a revised presentation of the current results necessitated changes in the numbering as well as the order of figures.

  1. The authors should have assayed the differentiation of iPSC into cardiomyocytes by revealing the expression of specific biomarker with fluorescent microscopy. Without this initial control, there is no certainty that the cytotoxicity of doxorubicin was investigated on cardiomyocytes.

The hiPCS-CM cells used in this work were acquired commercially (CDI, Madison, USA) and based on the supplier´s information we had not originally sought to verify the identity of the cells. Upon the mentioned comment, we have detected in these cells the presence of human troponin T and I as well as recoded their specific behavior, i.e. synchronized beating.

Expression of troponin T (TT) and troponin I (TI). Bar 20 µm, magnification 200x.

  1. The WST-1 assay used to measure cell viability wasn’t appropriate because it is known that doxorubicin affects mitochondrial functions and the assay is based on the production of formazan by a mitochondrial dehydrogenase. So the cell viability must have been quantified with another assay.

As based on the suggestion, we have determined cell viability in our work using neutral red uptake assay whose results we further validated with trypan blue assay. Results from NRU assay corresponded to the original results from WST-1 assay and were implemented into revised manuscript.

  1. There is no information in the manuscript about the method used for cell dead measurements.

Cell death rate was determined cytometrically as based on morphological analyses supported with data from nuclear staining. This information was added into revised manuscript.

  1. The authors should mention for each experiment, the N (or number of replicates).

It was done.

Reviewer 2 Report

In this article, the authors investigate the role of intracellular zinc in doxorubicin-induced cytotoxicity in hiPSC-CMs. The author used several phenotype analyses and indicated that zinc affected multiple aspects in iPSC-CMs.

Overall, I find the conclusions of this manuscript interesting and to be potentially impactful to the scientific community. Specifically, the role of Zn in cardio-oncology is not well known. Thus, this study provides a first exploration of this concept. However, some concerns should be addressed as listed below.

1.      Only one cell line is insufficient in iPSC study. More iPSC-CM lines may be needed (eg, n=2-3) to ensure the consistency of the conclusions since there is no in vivo experiments.

2.      The reviewer concerned about autophagy measurement. The gold standard test for autophagic flux is western blot/IF staining for MAP1LC3B/LC3B. it is still critical to validate the major conclusions using LC3 assay since its antibodies are easily accessible from commercial vendors. In particular, previous study have shown that doxorubicin inhibits autophagic flux in mouse heart (1), which is on the contrary with author’s conclusion.

3.      Rigor of this research is decreased by the absence of exact statistical information (eg, n number of each experiment) in the figure legends.

4.      The writing should be improved for a better understanding of the scientific content.

5.      Cell type-specificity of Zn's role should be tested. Given that Zn exist in various cell types (e.g., fibroblasts, cardiomyocytes and leukocytes). Is Zn important for cell survival in many different cell types? Or just for CMs?

References:

1.         Li DL, et al. (2016) Doxorubicin Blocks Cardiomyocyte Autophagic Flux by Inhibiting Lysosome Acidification. Circulation 133(17):1668-1687.

Author Response

Generally, we would like to thank the reviewer for her/his comments, constructive criticism and suggestions to improve the quality of our present work. All the raised questions and comments were taken into consideration and requested changes were implemented both as a part of originally existing evidence as well as based on the newly acquired data. Also, the implementation of new data as well as a revised presentation of the current results necessitated changes in the numbering as well as the order of figures.

  1. Only one cell line is insufficient in iPSC study. More iPSC-CM lines may be needed (eg, n=2 ,3) to ensure the consistency of the conclusions since there is no in vivo experiments.

We agree to this comment of the reviewer. Experiments with just one model – in this case cell line might not bring perhaps sufficient scientific robustness. We are well aware of such limitation and mentioned it in the discussion and conclusion. Still, given the time and the extent of experiments needed to recapitulate all reported findings we would estimate that a considerable time (perhaps even one year) would be required if other cell lines are to be used. We have already been informed by the editor of the journal that we had exceed given time for revisions (as based on editorial policy) and we might submit thiuus revised work of ours as a new submission. Therefore we believe that the final decision on this matter must be up to them.

  1. The reviewer concerned about autophagy measurement. The gold standard test for autophagic flux is western blot/IF staining for MAP1LC3B/LC3B. it is still critical to validate the major conclusions using LC3 assay since its antibodies are easily accessible from commercial vendors. In particular, previous study have shown that doxorubicin inhibits autophagic flux in mouse heart (1), which is on the contrary with author’s conclusion.

As based on reviewer´s comment and suggestion, we have employed dual fluorescent system Premo™ Autophagy Tandem Sensor RFP-GFP-LC3B Kit to evaluate autophagy flux in our model. Obtained results confirmed our findings regarding positive and not inhibitory effects of doxorubicin on autophagic flux in our model cells. We have added this evidence into our revised manuscript and attempted to discuss these results against suggested evidence.

  1. Rigor of this research is decreased by the absence of exact statistical information (eg, n number of each experiment) in the figure legends.

This information has been added as suggested.

  1. The writing should be improved for a better understanding of the scientific content.

We have modified writing to make it more understandable.

  1. Cell type-specificity of Zn's role should be tested. Given that Zn exist in various cell types (e.g., fibroblasts, cardiomyocytes and leukocytes). Is Zn important for cell survival in many different cell types? Or just for CMs?

Thank you for this comment and suggestion. Still, we are not exactly sure what is meant by it. Generally, as is commonly known (and stated in the text), Zn is vital for all cells, in particular given its involvement in a wide range of cellular functions and activities (extensive published evidence – for instance Metallomics. 2019 11(8):1330-1343, Mutat Res. 2012 733(1-2):111-21)

. However, its importance in particular cells is underscored by an absolute or relative content which differs. There are cells which contain more Zn and appear to be more dependent on its adequate management than others; these include for instance prostate cells, skin cells, some immune system cells, respiratory epithelial cells and heart cells. This extends to tissue or organ levels as well – prime example is neuronal tissue. Our study was aimed at just one type of cells in the heart – cardiomyocytes and did not include other cells present the heart. From this point of view, we feel the raised question is relevant since it is not known whether similar or different biological relationship reported here would apply for them or not. This might be the next stage, to employ more sophisticated model (i.e. organoid) to verify it.

Round 2

Reviewer 1 Report

The new version of the manuscript submitted by Rudolf K. and Rudolf E., regarding cell effect of doxorubicin on cardiomyocytes derived from human induced pluripotent stem cells (hiPSC-CMs), contains all the changes and precisions requested previously. 

More particularly, the measurements of cell viability were performed with a suitable assay. And also, the two minor points were rectified. 

Author Response

The new version of the manuscript submitted by Rudolf K. and Rudolf E., regarding cell effect of doxorubicin on cardiomyocytes derived from human induced pluripotent stem cells (hiPSC-CMs), contains all the changes and precisions requested previously. 

More particularly, the measurements of cell viability were performed with a suitable assay. And also, the two minor points were rectified. 

Authors thank the reviewer for the approval of revisions and changes of our manuscript as originally requested. Moreover, authors proofread the text again to correct spelling mistakes and typographical errors and to make small revisions in the text presentation.

Reviewer 2 Report

The authors addressed the comments from this reviewer. I do not have further comment to report.

Author Response

The authors addressed the comments from this reviewer. I do not have further comment to report.

Authors thank the reviewer for the approval of revisions and changes of our manuscript as originally requested. Moreover, authors proofread the text again to correct spelling mistakes and typographical errors and to make small revisions in the text presentation.